# Opinions of Older Individuals on Advance Care Planning and Factors Affecting Their Views: A Systematic Review

**DOI:** 10.3390/ijerph20105780

**Published:** 2023-05-10

**Authors:** Nilufer Korkmaz Yaylagul, Fatma Banu Demirdas, Pedro Melo, Rosa Silva

**Affiliations:** 1Department of Gerontology, Faculty of Health Sciences, University of Akdeniz, Antalya 07070, Turkey; 2Department of Gerontology, Faculty of Health Sciences, University of Turgut Ozal, Malatya 44210, Turkey; 3Institute of Health Sciences, Universidade Católica Portuguesa, 4200-450 Porto, Portugal; 4Centre for Interdisciplinary Research in Health, Universidade Católica Portuguesa, 4200-450 Porto, Portugal; 5Center for Health Technology and Services Research (CINTESIS), 4200-450 Porto, Portugal; 6Porto Nursing School (ESEP), 4200-450 Porto, Portugal; 7Portugal Center for Evidence Based Practice, A JBI Center of Excellence (PCEBP), 3030 Coimbra, Portugal

**Keywords:** supportive care, terminal care, end-of-life care, education and training, public health

## Abstract

The objective of this systematic review is to present older individuals’ views on the advance care planning (ACP) process and the factors affecting those opinions. The review contains search terms predetermined in the databases of CINAHL, MEDLINE (via PubMed), Academic Search Ultimate, Web of Science, Master FILE, and TR Dizin over the last 10 years (1 January 2012–31 December 2021) in English and Turkish. The studies were included in the research using inclusion (sample age ≥ 50, focusing on individuals’ opinions on ACP) and exclusion (articles whose samples consisted of individuals with a specific disease, non-research articles) criteria. Quality assessment was conducted using the Mixed Methods Appraisal Tool. A narrative synthesis was used to collate findings. The most striking results are the positive perspectives increasing in parallel with the individuals’ level of knowledge and experience about ACP. Variables affecting their views are advanced age, marital status, socioeconomic status, perception of remaining life expectancy, self-perceived health, number and stage of chronic diseases, religion, and cultural characteristics. This study offers guidance on the application and dissemination of ACP, empowering the use of this practice given the perspectives of older adults on ACP and the factors that affect them that the data show.

## 1. Introduction

Older individuals are at greater risk of chronic disease, physical dependence, and cognitive decline compared to other age groups, which puts them at risk for frailty syndrome. As a result of technological developments in modern medicine, practices (e.g., percutaneous endoscopic gastrostomy) that can prolong the life of individuals who cannot be treated and have life-threatening risks have emerged. However, these practices often ignore the emotional needs of the individual and their family and end up creating a significant challenge for them [1]. ACP is quite crucial at this point. ACP “enables individuals who have decisional capacity to identify their values, to reflect upon the meanings and consequences of serious illness scenarios, to define goals and preferences for future medical treatment and care, and to discuss these with family and health care providers. ACP addresses individuals’ concerns across the physical, psychological, social, and spiritual domains” [2]. In cases where older individuals lose the ability to communicate and make informed decisions about medical intervention due to chronic diseases and their consequent acute complications, the general and legal assumption is that the individual will choose to receive all necessary medical care to survive [3]. These care interventions, which only serve to prolong life, reduce the possibility of older individuals dying consciously, autonomously, and with dignity, and turn death into a technical phenomenon. Nevertheless, if a person specifies his or her future life-sustaining treatment and care preferences in advance while still competent and aware, this assumption is no longer valid, and the person’s expressed wishes guide medical decisions. Due to the increase in the older population and individuals with chronic diseases in society, it is crucial to empower these populations to request advance care planning (ACP). Thus, ACP has become a public health concern. Fried and Drickamer [4] emphasize the necessity of spreading awareness of ACP in the whole society.

Patient autonomy was first recognized under the “California Natural Death Act” in the United States (US) in 1976, which provided for those who elected a substitute decision-maker or documented their medical request in advance before losing their capacity to make decisions [5]. Later, in 1990, the US Congress adopted the “Patient Self-Determination Act” (ACP-Advance Care Planning) to encourage the use of advance directives (AD) [6]. This law mandated federal recognition of patients’ rights to refuse or accept treatment. Today, ACP is practiced in many countries, such as the US, Canada, the Netherlands, Australia, and China [7]. “Elements that may be included in an AD are an individual’s preferences regarding the use of life-sustaining technologies such as dialysis or ventilator machines, Cardiopulmonary Resuscitation (CPR), artificial nutrition, and palliative care” [8] (p. 304). ACP is a formalized communication process that allows individuals to discuss their goals and preferences for their future medical treatment and care, consult such goals and preferences with their families and healthcare providers, and record and review their final preferences at will. This process has both formal and informal components. Formal components include providing documentation of advance directives stating the person’s prior medical decisions regarding health care. Informal components, on the other hand, are discussions with family members and/or healthcare providers about one’s own end-of-life (EOL) care choices [9].

Studies on the impact of ACP on EOL care suggest that ACP reduces life-sustaining treatment, increases palliative and nursing home use, and prevents hospitalizations [10,11]. In addition, ACP has been found to improve compliance with the patient’s EOL wishes, increase patients’ and their families’ satisfaction with care, and reduce family stress, anxiety, and depression [12]. Despite all its benefits, participation in ACP is not at the desired level, even in legally supported countries [13]. Public health approaches aim to provide interventions that encourage discussion of ACP in a way that includes the whole society, not just individuals who are clinical and health care recipients. To expand participation in ACP, the identification of factors affecting participation decisions and the reasons guiding future interventions is required. Systematic studies on the views of individuals about ACP and the factors affecting them are often not comprehensive and focus on specific groups, such as terminally ill patients, aged Australian individuals, etc. [14,15]. This study aims to present the opinions of older individuals on ACP and the factors affecting those opinions comprehensively. Thus, this systematic review aims to answer the following questions:How do older individuals approach the ACP process?What are the factors that affect individuals’ views of ACP?

## 2. Materials and Methods

A systematic review was conducted to provide a comprehensive and unbiased synthesis of relevant studies in a single document [16]. 

### 2.1. Literature Review Strategy and Eligibility Criteria

A wide range of social and health databases was used, including CINAHL, MEDLINE (PubMed), Academic Search Ultimate, Web of Science, Master File, and TR Dizin, to undertake a broad review of the research topic and obtain all the relevant current research articles. The Google Scholar website was searched with the same keywords to obtain different papers.

Terms with similar meanings in the literature review were set to include “Advance care planning”, “End-of-life planning”, “Advance Directive”, “End-of-life decision”, “Advance health directive”, “elderly”, “aged”, “older”, “elder”, “elderly people”, “old people”, “older individual”, “senior”, “attitudes”, “perception”, “opinions”, “thoughts”, “feelings”, and “beliefs” for searching within the title and abstract. Terms and descriptors and their combination were used dependently according to the sensitivity and specificity of each database used. 

### 2.2. Inclusion and Exclusion Criteria

Inclusion criteria were established as follows: English and Turkish research articles including participants 50 and over, as the rate of death and chronic diseases has been found to be high in several studies [17,18], and studies focusing on participants’ opinions on ACP, published between 2012 and 2021 (because we believe that in the last 10 years, enough research has been carried out to meet the objectives of this review), with full text access. Exclusion criteria were applied as follows: research articles whose samples consist of individuals with a specific disease or where the age of the sample is under 50; research articles not in English or Turkish; and systematic review or review articles, books, book chapters, abstracts, and reports and editorials.

### 2.3. Review Process

The Preferred Reporting Items for Systematic Reviews and Meta-Analysis (PRISMA) protocol was used to identify relevant articles in the study [19]. The flow diagram in Figure 1 provides an overview of the review and selection procedures.

A total of 208 articles was obtained. After the removal of duplicate articles and re-evaluation based on the inclusion and exclusion criteria, 34 articles were retrieved. Afterward, the titles and abstracts were read separately by the researchers (NKY, FBD). Full texts were retrieved from the system based on eligibility. Selected articles were compared by the researchers (FBD, RS, PM), and a consensus was reached. Finally, 17 articles were included in this review. 

The quality of all studies included in the research was evaluated with the Mixed Methods Appraisal Tool—MMAT (Testing the reliability and efficiency of the pilot Mixed Methods Appraisal Tool—MMAT), version 2018, developed by Pluye and Hong and applicable to qualitative, quantitative (randomized, non-randomized, and descriptive) and mixed methods study designs [20]. The MMAT is based on answering “yes/no” to a series of questions to determine the quality of the study design. A score of “1” is received for each yes answer and “0” for each no answer. The total score gives the MMAT score of the study under consideration. Researchers, in pairs, evaluated and compared the studies separately using this tool. As a result of the evaluation, and based on quality, no studies were excluded. Quality evaluations are shown in Table 1.

### 2.4. Data Analysis and Synthesis

Narrative synthesis was used in this study as the selected articles were not homogeneous and consisted of qualitative, quantitative, and mixed studies. Reference and study year, aim, MMAT score, sample, methodology, and main results were extracted from the data, and a Data Assessment Form (DAF) was created for the textural description (FBD) (Table 1). The studies’ results were analyzed using a coding scheme. Data coding was performed by the researchers (FBD, NKY), and a consensus was reached. In this respect, thematic analysis was carried out for qualitative studies. The themes identified were interest and sensitivity for ACP; timing; family roles; difficulty in discussing end of life; traditional and religious values; unreliability regarding the uncertainty of end of life; and ACP experience, knowledge, and awareness. The factors affecting the ACP decisions in quantitative studies were evaluated, and demographic characteristics and ACP knowledge level, life experiences, religiosity and death attitudes, chronic diseases, and future life perspective factors were obtained.

## 3. Results

### 3.1. General Characteristics of the Studies

The general characteristics of the articles included in the research are presented in Table 1. Accordingly, nine of the reviewed articles were quantitative, six were qualitative, and two were mixed-method studies. In terms of publication years, most studies (*n* = 4) were conducted in 2019, and three in 2020. Considering the countries in which the studies were performed, the USA had the highest rate with seven studies, followed by China with three studies. Other countries with one study each were Korea, Japan, the Netherlands, Turkey, Taiwan, Canada, and Australia.

### 3.2. Views of Individuals on the ACP Process

Based on the first research question, “What are the views of individuals on the ACP process?”, the following common themes were extracted from the thematic analysis of six qualitative and two mixed studies.

#### 3.2.1. Interest and Sensitivity for ACP

Statements under this theme were obtained from three studies [11,31,33]. In this context, it is understood that the participants expect attention from the physicians and expect them to initiate ACP discussions and provide information [31,33]. However, they complained that the physicians did not spend enough time on these discussions and were not interested, due to time pressure and the difficulty of talking about such topics. Some of the participant’s statements that best reflect this are as follows: “Because I don’t think it means anything to him (doctor)… That’s up to you’ or so, he would say…”, “He (doctor) does not have time for it.”, “No, I mean… I feel sorry for those people… they have this enormous time pressure if you ask me… That’s very unfortunate…, yes.” [11] (p. 521).

#### 3.2.2. Timing

Most of the studies emphasize the importance of the timing and place for the ACP discussions [11,22,24,31,33]. Accordingly, healthy and active individuals delay the initiation of ACP discussions [22,24,33]; in addition, holding such discussions too early may cause anxiety for the individual and family members [22]. In the focus group study by Yonashiro-Cho et al. [22] (p. 1887), one of the participants described this situation as: “I think when nothing happens yet, there is no need to talk to our family about anything. Not until the doctor tells us that we have an incurable disease should we talk to our family about the arrangements or our wishes. Otherwise, they will think there is something very wrong with us”. In the qualitative study by Glaudemans et al. [11] (p. 522), the subjects stated that making decisions at an old age would not be healthy, and therefore the decisions should be left to the younger family members. One of the most relevant statements is: “Ten years ago I would make decisions more calm and well balanced than I would now… So, if we old folks have to refer to each other, will that be safe? I wonder. I think we’d then better burden the younger generation”. The study by Ko and Nelson-Becker [31] with older people who were homeless and living in temporary housing suggested that such decisions were not a priority for them.

#### 3.2.3. Family Roles and Difficulty in Discussing End of Life

Apart from the study by Ko and Nelson-Becker [31], all qualitative and mixed studies highlight the role of the family in the ACP process and discussions. While some individuals stated that they approached ACP positively as they did not want to increase the care burden on their families [24,27,32] others said that the decision should be made by their families [11,32], and therefore it is necessary to discuss these issues among the family beforehand [33]. The statement of one of the 16 participants in the qualitative interview in the mixed study by Zhu et al. [24] (p. 747) demonstrates this view well: “My daughter will help me in decision-making at the end of life. I might go to extremes, and it is selfish to make my own decision”.

The study by Yang et al. [26], on the other hand, revealed that although older adults would like to have EOL care discussions with their family members, these members avoided such conversations. The study by Yap et al. [27] revealed that the participants thought that the individual should make these decisions autonomously, independently from the family, and that talking about EOL issues with the family would worry family members. In both studies, the participants mentioned the difficulties of talking about EOL and ACP, and therefore the studies drew attention to the informal or indirect ways of expression [26,27].

#### 3.2.4. Traditional and Religious Values

Studies reveal that traditional and religious values have an impact on opinions about ACP [24,27,31,33]. Two studies reveal that it is uncomfortable or ominous to talk about death-related issues, and two studies report that it should be left to God and no such planning is required [31,33]. The study by Yap et al. also proposes that not talking about or discussing such issues is related to tradition. A participant’s statement is as follows: “Some people would probably mind talking about such topics, as it might be deemed as a taboo… The introduction of ACP in these people would be difficult. These people would probably prefer their family members to make EOL care plans for them” [27] (p. 3303).

#### 3.2.5. Unreliability Regarding the Uncertainty of End of Life

According to the results of the three studies, participants emphasized that it was hard to have such conversations and make decisions at that moment since the individual’s future health conditions were uncertain and variable, and these early decisions may be different in the future [11,24,32]. Individuals believe that it is not easy to make decisions about the end of their lives and this has been an obstacle to the initiation of ACP discussions. Two statements, by Glaudemans et al. and Fan et al. [11,32], describe the difficulty of predicting the participants’ future preferences, respectively, as: “It does not make a difference if we talk about it, because when it comes to it, I wonder what I would have wanted a year before… if that is still the same… it could just as well be something different” [11] (p. 521). “How can I know everything? If there was a new treatment or something that happened accidently, would I need to make decisions for all possible conditions?” [32] (p. 3455).

#### 3.2.6. ACP Experience

Statements under the theme of having ACP or EOL experience were found in four studies [11,26,32,33]. Fan et al. [32] report in their study that an individual’s experience with ACP facilitates the expression of views by imagining clinical situations. Other statements suggest that an individual’s exposure to life-prolonging care practices indicates a more positive view of participation in the ACP [11,26,33]. One participant’s statement regarding life-prolonging care practices is: “When my wife died, my son said that he would save his mother anyway. In fact, I do not want to save her. The thick pipe was stuck in my wife’s body, and she must be in pain. One day, I will make it clear to my son that I do not want to suffer” [26] (p. 183).

#### 3.2.7. Knowledge and Awareness

The lack of knowledge and awareness of individuals emerges as one of the most important reported reasons for not having a prior directive [22,24,26,27,33,35]. Two studies included critical comments reflecting the participants’ lack of knowledge about how the ACP is performed, what it encompasses, and the process and protocol for filling out the forms [22,33]. In two studies, participants showed interest in the subject even though they had no previous knowledge of ACP [24,26]. Yap et al. [27] determined the factors affecting the participation in ACP in a study with 30 Chinese Australian participants aged 55 and over and found out that individuals were informed about ACP through their environment, and they were not able to fully understand or experienced conceptual confusion as they did not have full command of the spoken language. Zhu et al. argue that insufficient information about ACP and EOL treatments in hospitals causes individuals to insist on continuing traditional treatment. One of the participants in this study stated in this respect: “When family members were in a hospital, doctors would not discuss with us other than the condition, we had to adopt the most traditional treatment method, which was to insist on curative treatment” [24] (p. 746).

### 3.3. What Are the Factors That Affect Views of ACP?

The results of nine quantitative and two mixed-method studies were analyzed based on the question ‘What are the factors that affect views of ACP?’

#### 3.3.1. Demographic Characteristics and ACP Knowledge Level

The percentage of female participants in studies except five (one study female = male, four studies male > female) was higher (%70.6) than that of males. There are seven studies [22,23,25,26,28,29,31] that determine socioeconomic status using markers (income status, regular income, presence of health insurance, whether aid is received, etc.) and address ACP-related behavior, six studies [18,22,24,25,27,30] that examine racial and ethnic differences, and nine studies [11,17,23,24,26,27,31,32,34] that address religious differences and being religious/non-religious.

Zhu et al. [24] found that respondents living alone or living with family members tended to discuss ACP more frequently if compared with respondents living with someone else (friends, carers, etc.) (*p* = 0.003), and the group with higher socioeconomic level was found to approach the discussions more positively (*p* = 0.040).

In their study of individuals with chronic diseases, Yang et al. [26] found that those with higher monthly income and educational level and those unmarried (*p* < 0.001) had higher ACP acceptance (*p* < 0.001) The results of the study by Kwak et al. [18] on American Indians and non-Hispanic White Americans suggest that American Indians were significantly less likely than the other group to have an EOL care plan, to have a durable power of attorney for healthcare (DPAHC), and to complete an ACP. Multivariate logistic regression results showed that having an EOL plan was associated with old age (95% confidence interval (CI) = 1.02, 1.06), having a college degree or higher (95% CI = 1.01, 3.95), and having more chronic conditions (95% CI = 1.11, 1.39), but not with race.

Quantitative studies also show that familiarity with definitions of ACP plays a determining role in ACP behaviors, similar to qualitative studies. Zhu et al. [24] found out that among 523 participants aged 60 and over living in China, only 5% could define ACP correctly and most participants (92.7%) were not familiar with ACP and ACP-related terminology, only 16.1% of the participants had knowledge of living will, and less than 10% were able to explain the meaning of proxy decision-maker or EOL decision-making.

Similarly, Zhang et al. [23] found that among 900 Chinese older adults, 78.3% had never heard of ACP. ACP knowledge was directly related to educational level. In the same study, 80.9% of the participants stated that they expected true information about their health status from their doctors, 52.4% said that they wanted to make their own health decisions, 55.8% indicated that they did not want life-prolonging treatments in irreversible cases (higher in those under the age of 70 than in those over 70), and 39.4% indicated that they would like to fill out an ACP document in case of a serious illness (higher for those with higher educational level).

Sahin and Buken [29] interviewed 448 participants aged 65 and over and as a result, 72.5% of the participants stated that they should be informed about the medical diagnosis, and 70.1% stated that the decision to inform the family should be left to the patient, and in any case, the physician should inform the patient. The study showed that long-term acquaintance with the physician was critical in fulfilling the patient’s wishes, and 79% of the participants would consider making a will and assigning a medical guardian.

The study by Tripken et al. [25] on 77 participants aged 55 and over residing in two socioeconomically different neighborhoods (classified as low and high) showed no significant difference between the two communities in terms of beliefs and attitudes toward end of life. Nevertheless, there were significant differences in familiarity with and knowledge of ACP terminology between the two communities, as well as in the completion of further directives. High socioeconomic levels positively affected ACP knowledge (*p* = 0.00), and the participants in this community were more likely to participate in ACP. The education level was higher in high-income facilities, and there was a significant difference between the two groups (*p* = 0.00). Considering the relationship between educational level and knowledge, it may also explain why 95% of the residents in high-income facilities had an ACP, whereas only 54% of those had it in low-income facilities.

#### 3.3.2. Life Experiences

Carr [36] examined the relationship between participants’ experience of the death of close relatives and their completion of an ACP; it has been stated that individuals who have lost their loved ones traumatically considered their own end more consistently and had higher motivation to join ACP. In the study by Miyashita et al. [28], participants with experience of being with dying family members were more likely to engage in ACP discussion than those without (fully adjusted prevalence ratio 1.31, 95% CI 1.04–1.65). The study by Zhu et al. [24] with 523 participants found that individuals who experienced a family member or friend being exposed to a life-supporting intervention (LST) (*n* = 300, 57% of participants) were more likely to view ACP positively than others (*p* < 0.001).

The study by Amjad et al. [21], which included 304 participants aged 60 and over, evaluated the relationship between EOL care or decision-making experiences and four different ACP acceptance behaviors (no intention to change behavior in the near future/consideration of change/preparation/action). It was concluded that older individuals with EOL care experience or decision-making experience for others were keener on participating in behaviors related to ACP.

#### 3.3.3. Religiosity and Death Attitudes

Zhu et al. [24] evaluated the relationship between religiosity and ACP and showed that those who defined themselves as religious (*n* = 109/20.8%) tended to be more willing to discuss ACP in the future (*p* = 0.026). Dobbs et al. compared the levels of religiosity and death attitudes of 157 participants with chronic diseases and three different stages of ACP discussions (discussions with doctors, discussions with family members, and completion of a will related to the end of life) and found that the closer the participants felt to God (religiosity), the more likely (5.26 times) they were to have ACP discussions with doctors (95% CI = 19.22, *p* < 0.05) [17]. For each unit increase in participants’ mean fear of death score (0.66), they became 34% less likely (95% CI = 0.47, 0.94, *p* < 0.05) to complete an ACP. This finding is consistent with other studies, suggesting that fear of death behaviors, such as denial of death, interfere with planning for EOL and the death process [37,38].

#### 3.3.4. Chronic Diseases and Future Life Perspective

Studies investigating chronic diseases and related factors (Future Time Perspective, Scale for assessing the patient’s ability to perform physical activities of daily living) present different results regarding ACP. The study by Amjad et al. [21] showed that having a severe illness or undergoing major surgery was not associated with ACP behavior.

Respondents with more depressive symptoms were less likely to complete a durable power of attorney for health care (DPAHC) (95% CI = 0.86, 1.00), and having more chronic conditions was associated with a higher probability of completing a DPAHC (95% CI = 1.08, 1.35) according to Kwak et al. [18].

Yang et al. [26] evaluated 471 individuals aged 60 and over with chronic diseases and found out that the higher the score on the scale of 10 personal activities (controlling bowel, controlling bladder, grooming, getting on and off the toilet, feeding, mobility, exercising, dressing, bathing, and using stairs, with scores being identified between 0–3, 0 = unable/dependent to 3 = independent), the higher the acceptance of ACP (*p* < 0.001).

Kim et al. [34] investigated 112 Korean people aged 65 and over with low socioeconomic status and chronic diseases and showed that married participants preferred more aggressive EOL treatments [CPR (Cardiopulmonary resuscitation) 26.2% vs. 7.1%, *p* = 0.005; ventilation support 21.4% vs. 2.9%, *p* = 0.001 and hemodialysis 16.7% versus 4.3%, *p* = 0.026] compared to single participants.

Future time perspective (FTP) can be expressed as a person’s perception of their distance from death. FTP influences the goals people set for themselves and the social interactions they intend to pursue. It assumes that perceiving future time as interrupted or finite leads to goals and social relationships that provide immediate emotional rewards [30]. Luth’s study, which was based on this theory, used its data from the New Jersey EOL study and investigated the extent to which cognitive processes, especially perceptions of one’s distance from death, are related to ACP, and found a correlation between those with expansive (individuals who think they will be alive in the next 10 years for sure) FTPs and ACP participation. The ACP participation rate of people with expansive FTP was compared to that of those with moderate (individuals who think they will be alive in the next 5 years but are not sure they will be alive in the next 10 years) FTP, and the former were 58% less likely to discuss EOL (*p* = 0.011). Similarly, having an expansive FTP reduced the probability of participating in the official ACP by 70% (*p* < 0.00) [30].

## 4. Discussion

This systematic review aims to reveal the views of individuals about ACP and the factors affecting such opinions. The results of the studies show that the lack of knowledge and awareness of older adults plays a critical role in their participation in ACP-related processes. Studies addressing the lack of knowledge about ACP suggest that high socioeconomic status and having a higher education degree have a positive relationship with ACP knowledge level [25]. On the other hand, the sources of the lack of knowledge regarding the application procedures of the ACP were investigated in some studies [22,33]. The reasons can be listed as follows: (i) not learning information about one’s own health from trusted experts (especially physicians) [23]; (ii) the inability of physicians to initiate ACP discussions and provide consultancy services due to time constraints and work pressure, thus not allowing individuals to access sufficient and correct information [31,33]; (iii) confusion regarding concepts or difficulty understanding them in regions where culturally and linguistically diverse communities live [27]; and (iv) finally, hospitals providing insufficient information about ACP and EOL treatments. These reasons prevent individuals from evaluating the last phase of life and having the autonomy to make an alternative decision, causing them to continue traditional treatments and interventions [24]. Continuous counseling offered by physicians in centers where preventive and curative health services are provided can be a way to reach individuals with different characteristics.

Studies on the attitudes and actions of older adults regarding their participation in ACP processes and their characteristics suggest that a high socioeconomic level positively affects the level of ACP knowledge and indirectly increases the probability of participating in ACP [24,25]. Confirming this, Yang et al. [26] found that those with a higher monthly income and educational level also had higher ACP acceptance. In the study by Ko and Nelson-Becker [31] with homeless older adults, participants stated that such a decision was not a priority due to the challenging conditions they were under at the time, which clearly revealed the relationship between socioeconomic status and the ACP decision.

A study comparing the racial variable and individuals’ attitudes towards ACP showed that having a university or higher education degree, a higher number of chronic diseases, and advanced age were associated with ACP, however, race was not [18]. Although the racial variable appears to have no effect on ACP decisions, these results may emphasize how social resources are shaped by race and socioeconomic status, and the impact of this stratification on health outcomes [39].

Studies have found that unmarried people have higher acceptance of ACP [26] and that married individuals are more likely to prefer life-prolonging treatments, while older respondents living alone or with a spouse (and children) tend to discuss ACP in a more positive light than respondents living with someone else (friends, carers, etc.) [24].

Studies addressing the issue from a religious perspective propose that individuals who identify themselves as religious tend to be more willing to discuss ACP in the future [24] and that religious people are more likely to have ACP discussions with doctors [17]. These results show that the religious characteristics of the participants should be considered in the planning and initiation of ACP discussions. In addition, patients’ and/or their families’ cultural and/or religious backgrounds should be considered to understand their opinion on ACP.

Being healthy and active, as another variable, was considered a factor delaying the initiation of ACP discussions in the studies [24,33]. Studies suggest that participants with poorer health are more likely to positively approach ACP discussions than those in good health [24] and that having more chronic conditions is associated with a higher probability of completing DPAHC [18]. Having a severe illness or undergoing major surgery is not associated with stages for most ACP behaviors [21]. However, individuals with more depressive symptoms are less likely to complete a DPAHC [18]. Moreover, individuals with expansive FTP in terms of perceived distance to death or remaining life expectancy were found to be less likely to participate in ACP than those with moderate FTP (middle category) [30].

The increased fear of death is considered a factor that negatively affects ACP participation [17], and individuals’ attitudes toward death are another element that has an impact on how ACP is viewed. It has been observed that the belief that discussing the subject will bring bad luck [24,31] and that end-of-life planning should be left to God [31,33] has a negative impact on participation in ACP discussions. Accepting both life [27] and death [26] as they are facilitates the empowered discussion of ACP, which is an expected result considered in the cultural context.

Studies focusing on the ACP experience of the participants argue that having previous ACP experience is a critical factor that determines the individual’s perspective on ACP. Individuals with experience of being with dying family members [28], exposure to an experience of life-prolonging care practices in their environment/family, or making decisions on behalf of others for EOL care [21] have a more positive view of ACP participation and discussion [11,24,26,32,33].

Studies suggest that timing, family relationships, and confidence in the future are significant factors in individuals’ participation in ACP discussions and processes. Some studies have drawn attention to the fact that individuals may use informal or indirect ways of expression or should decide alone, as early discussions may cause anxiety in the individual or family members [22,27]. Others argue that an older adult may think of leaving these decisions to the younger members of the family [11,32], as the decisions made in old age will not be reliable and to avoid increasing the care burden on the family [33]. However, those who are willing to talk can also be hindered by their family members who do not display such willingness [26].

In this systematic review, knowledge and raised awareness emerged as noteworthy drivers in studies focusing on the factors affecting ACP decisions in old age. It is important to disseminate knowledge and awareness about ACP. The studies demonstrate that socioeconomic level, the timing of decisions, chronic diseases and loss of relatives, ACP experience, and a family’s approach to ACP are critical factors in the ACP decision-making process.

Empowering individuals to make better decisions regarding ACP is a function of all health professionals, including nurses, doctors, psychologists, and managerial specialists, among others. In addition, the empowerment of citizens, sick or not, is a valuable resource that allows for the creation of moments of information exchange and decision-making between citizens and health professionals, which helps to promote equity, where everyone is treated respectfully and on an equal footing according to their capabilities and needs. Knowing the factors that could influence the views of older people about ACP is an essential tool for planning the care and empowerment of these subjects [40].

## 5. Conclusions

According to older individuals’ views on ACP, we know that there are a number of factors that affect their views. Individual socio-demographic factors include socioeconomic status, self-perceived health, stage of chronic diseases, cultural characteristics, etc. There are also factors such as knowledge level and awareness of ACP that can be altered. ACP is an individual planning process, based on knowledge empowerment of the individual, i.e., of each citizen, which aims to improve the quality of life of dying patients, maintain human dignity, and protect medical resources in line with the individual’s directives. In addition to this, ACP can improve communication among patients, family members, and healthcare professionals, effectively reducing the decision-making burden of family members, and increasing the satisfaction of both patients and family members with healthcare professionals. Therefore, it is necessary to disseminate knowledge and awareness of ACP at national and international levels. In line with the research results, individual characteristics and experiences should be kept in mind in training and awareness studies for ACP. Efforts to increase public knowledge and awareness of ACP should consider individual socioeconomic status, educational status, chronic diseases, and religious beliefs. The success of public health interventions on ACP dissemination will be linked to evidence-based planning and implementation of interventions that normalize public discussion of this difficult issue.

Furthermore, ensuring the participation of all family members in ACP can also improve decision-making about this planning. It should be considered that advanced care planning requires a process that includes familiarity and information. It is possible for patients and family members not only to make a decision prior to any diagnosis but also to revise ACP when a condition arises. Therefore, it is crucial to understand barriers and facilitators for specific patients and their family members’ opinions on the ACP process during changing conditions.

The most important factor regarding the study’s limitations is its extent. In the study, the extent of the research was limited by searching only in two languages. We may have missed potentially interesting studies due to limiting the sample to articles published in English or Turkish.

An additional limitation of the review is time. In order to limit data in terms of our analytical ability and to access up-to-date research, a 10-year period was chosen. The inability to evaluate the studies prior to this period can also be considered an important limitation.

The most important strength of the research is that both qualitative and quantitative data on ACP were examined. Furthermore, the fact that both the views of individuals and the factors affecting these views are put on display in this study will contribute to the development of ACP.

## Figures and Tables

**Figure 1 ijerph-20-05780-f001:**
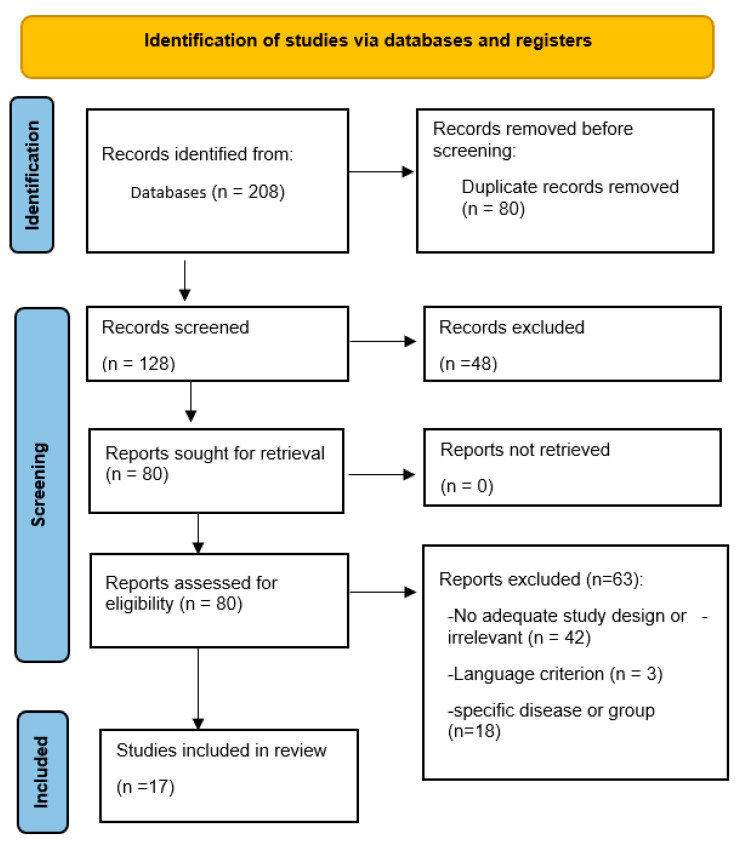
Flow diagram for systematic review process adapted from the PRISMA statement [19].

**Table 1 ijerph-20-05780-t001:** The results of the studies (*n* = 17).

Reference	Country	Aim	Score of MMAT	Method	Sample	Main Findings
Amjad et al., 2014 [21]	USA	To examine whether experiences with illness and EOL care are associated with readiness to participate in advance care planning (ACP).	4	Observational cohort study	Persons aged 60 and older recruited from physician offices and a senior center	Of 304 participants, 84% had one or more personal experiences (Life-threatening illness, risky or major surgery, life-threatening illness, or surgery) or experience with others (Medical decision for someone who died, known someone who had a bad death due to receiving too much medical care, known someone who had a bad death due to receiving too little medical care, experienced the death of a loved one who made wishes about end-of-life care known). Personal experiences were not associated with greater readiness for most ACP behaviors. In contrast, older individuals who have experience with EOL care of others demonstrate greater readiness to participate in ACP. Discussions with older adults regarding these experiences may be a useful tool in promoting ACP.
Yonashiro et al., 2016 [22]	USA	To explore Chinese-American older adult beliefs about and attitudes toward ACP, including attitudes toward engaging in formal and informal communication of care preferences with family members and healthcare providers	5	Qualitative, focus group, descriptive, relational	Volunteer participants among individuals receiving services from a large, community-based Chinese-American social work organization	Results showed that knowledge and experience of ACP and EOL decision-making varied by focus group, and few participants had prior instruction. It has also been observed that Chinese older adults prefer to use informal contexts rather than a formal meeting to convey their care preferences to loved ones and indirect communication strategies such as commenting on the circumstances of others rather than expressing their wishes directly.
Zhang et al., 2015 [23]	China	To investigate the preferences of ACP and healthcare autonomy in community-dwelling older Chinese adults	4	Quantitative, descriptive	Volunteer participants in two communities in Beijing’s Chaoyang District	A low level of “advance planning” awareness and preference was evident, although the majority of community-dwelling older Chinese adults appeared to have healthcare autonomy and rejected life-prolonging interventions in terms of EOL care.
Zhu et al., 2020 [24]	China	To measure the ACP-related knowledge and attitudes of Chinese community-dwelling older adults	5	Mixed-method, explanatory	Randomly selected volunteers from four randomly selected communities in Zhengzhou who met the inclusion criteria	The result of the study showed that the awareness and participation in ACP by older adults living in the community in mainland China are insufficient.
Glaudemans et al., 2020 [11]	The Netherlands	To explore older people’s and their families’ experiences with ACP in primary care.	5	Qualitative, descriptive	Elderly individuals and their families who GPs and nurses identified by the snowball method through national and regional service providers of aged and palliative care and who have had ACP interviews in the last three months	As a result, three main themes were identified: “openness and trust”, “timing and issues” and “family roles”. Participants were more receptive to ACP if they wanted to prevent certain future conditions and were less open to general practitioners (GPs) if they were insecure or had negative opinions about the time and attention they devote to ACP. The quality of ACP seems to improve if one discusses the participants’ views on their current life and future, a few specific future care scenarios, and the expectations and responsibilities for the ACP. It has been observed that the quality of the ACP increases if the family is involved in ACP discussions.
Tripken et al., 2019 [25]	USA	To assess the knowledge, attitudes, and beliefs about ACP among older adults in two socioeconomically diverse settings to identify the individual and contextual factors that influence behaviors regarding EOL care.	5	Quantitative-relational	Volunteer participants living in two non-profit independent living facilities (reflecting different socioeconomic populations) in a large mid-Atlantic suburban town	As a result of the research, significant differences were found between the two communities in their familiarity and knowledge of ACP terminology, as well as in the completion of advanced instructions and communication. No difference was found in attitudes and beliefs about EOL problems.
Yang et al., 2021 [26]	China	To investigate acceptance and influencing factors of ACP for community-dwelling elderly patients with chronic diseases in the Republic of China.	5	Mixed-method, relational	Volunteers selected through facilitated sampling from individuals (who met the inclusion criteria) living in three communities in Jinzhou City, Linhe District, Guta District, and Taihe District, China, Liaoning Province	The results of the study revealed that attitudes towards death and quality of life are the main determinants for the selection of ACP, and family support and past medical experience are also important factors.
Yap et al., 2018 [27]	Australia	To identify factors that influence the engagement of Chinese Australians with ACP.	5	Qualitative, descriptive	Participants are volunteer Australians of Chinese descent who speak Mandarin or Cantonese (who meet the inclusion criteria), purposively selected from five different communities in the Melbourne metropolitan area.	Three main themes were identified in the study: knowledge, attitudes, and needs regarding ACP. Low awareness and confusion about ACP were found among the participants. Although most participants reported positive attitudes towards ACP, they acknowledged that others might be uncomfortable discussing death-related issues. The participants reported that they would like to plan ahead in consultation with their family members to know their real health status and to reduce the burden on their families and the suffering for themselves. Language has been identified as the biggest hurdle to overcome in order to raise awareness of ACP.
Miyashita et al., 2021 [28]	Japan	To examine the associations between experiences of being with a dying family member and ACP discussions among Japanese older adults.	5	Quantitative-relational	Participants aged 65 and over who attended internal medicine, gastroenterology, cardiology, endocrinology, pulmonology, or orthopedics outpatient clinics at Shirakawa Kosei General Hospital in Fukushima, Japan, within a one-week period and answered a self-administered questionnaire	As a result, 32% of the participants were found to have an experience of being with their dying family members. Respondents with the aforementioned experiences were found to be more likely to engage in ACP discussions than those without them.
Sahin & Buken, 2020 [29]	Turkey	To determine the opinions of the elderly about end-of-life (EOL) decisions, which will act as a guide for physicians and families in clinical decision-making processes.	5	Quantitativedescriptive	The population of the study consists of elderly individuals (who meet the inclusion criteria) living in the city center of Burdur, Turkey.	As a result of the research, 72.5% of the participants stated that they should be informed about the medical diagnosis, 70.1% stated that the decision to inform the family should be left to the patient, and in any case, the physician should inform the patient. In addition, it was determined that it is important to know the physician for a long time to fulfill the patient’s wishes, and 79% of the participants could consider leaving a will and appointing a medical guardian.
Kwak et al., 2019 [18]	USA	To identify different views regarding ACP between older American Indian individuals and whites.	5	Quantitative-relational	American Indian and white elderly respondents living in Minnesota and South Dakota (who met the inclusion criteria)	The results of the study found that American Indian older adults were significantly less likely than their white peers to have an EOL care plan, durable power of attorney for healthcare (DPAHC), or a living will. Multivariate logistic regression showed that having an EOL plan was associated with advanced age, having a bachelor’s degree or higher, and having a greater number of chronic conditions, but was not associated with race. Having a DPAHC is associated with being white, being older, having lower levels of depressive symptoms, and having a greater number of chronic conditions, while completing a living will is associated with being white, being older, having a college education, and having more chronic conditions.
Luth, 2016 [30]	USA	To explore the extent to which cognitive processes, specifically perceptions of one’s distance to death, are associated with informal and formal ACP in a sample of older adults	5	Quantitative-relational	The study used data from the 2006–2008 New Jersey EOL Survey of 305 non-institutional adults aged 55–91 seeking care at one of three New Jersey medical centers.	Research results showed that people who perceive their remaining life expectancy to be large or limited are less likely to formally plan for EOL compared to those in the intermediate category. The relationship between future time perspective (FTP) and EOL discussions was not statistically significant.
Ko & Nelson-Becker, 2014 [31]	USA	To explore perspectives, needs, and concerns relating to ACP among older homeless adults.	5	Qualitative, descriptive	Volunteer older adults residing in a temporary housing facility in an urban area on the west coast of the USA	The main themes that emerged in the research were “disturbing the subject”, “trust in God’s decisions”, “physicians are preferred as decision makers” and “planning is important but not an immediate concern”. In addition, it was determined that it is desirable to show sensitivity to the homeless.
Dobbs et al., 2012 [17]	USA	To examine the association of religiosity and death attitudes with self-reported ACP in chronically ill older adults.	5	Quantitative-relational	Volunteer older adults with chronic disease that fit the inclusion criteria in primary care clinics in North Carolina.	According to the results of the study, more reported religiosity and physician-reported ACP discussions were found to be significantly related. It was also revealed that decreased fear of death was significantly associated with the completion of a will.
Fan et al., 2019 [32]	Taiwan	To explore the experiences and processes of ACP discussions in older residents of a long-term care institution.	5	Qualitative, descriptive	Participants are healthy volunteer adults aged 65 and older living in a long-termcare institution in eastern Taiwan.	As a result of the research, it was determined that the ACP process included not only personal ideas about good death for the participants but also the concerns of their families.In addition, uncertainty regarding medical and legal questions was identified as a barrier to decision-making in the ACP process. It was determined that on account of the participants perceiving themselves as a burden to their family members played an important role in this process. As a result of this, it was observed that during the ACP process, the participants took into account the opinions of family members or asked family members to make decisions for them.
Simon et al., 2015 [33]	Canada	To explore seriously ill, older hospitalized patients’ and their family members’ perspectives on the barriers and facilitators of ACP.	5	Qualitative, descriptive	The study used data from former studies conducted in 12 Canadian acute care hospitals.	As a result, three themes were identified that contribute to whether patients and family members participate in ACP. The resulting themes are “person” (patient or family responder characteristics), “access to physicians and ACP resources” and “doctor-patient/family interaction”.
Kim et al., 2019 [34]	Korea	To examine the relationship between some modifiable (i.e., perceived benefits/barriers and knowledge) factors and different advanced directive (AD) treatments in community-dwelling, low-income, older adults with chronic diseases.	4	Quantitative-relational	Individuals with low-income and scope-limited chronic diseases receiving home health services in Korea	As a result, it was observed that approximately half of the participants preferred hospice care (54.5%). Being married was associated with higher desires regarding cardiopulmonary resuscitation (CPR) and ventilation support, while having higher education and cardiovascular disease were associated with a preference for CPR and hemodialysis. It was found that modifiable factors such as perceived benefits/barriers and knowledge were associated with different types of AD treatments. Greater perceived barriers increased the likelihood of CPR preference but decreased the likelihood of hospice care preference. Greater perceived benefits decreased the likelihood of CPR preference and ventilation support, and AD knowledge decreased the likelihood of hemodialysis preference

Legend: ACP: advance care planning; AD: advanced directive; CPR: cardiopulmonary resuscitation; EOL—End of Life; DPAHC: durable power of attorney for healthcare; FTP: future time perspective; MMAT: Mixed Methods Appraisal Tool.

## Data Availability

For data supporting reported results please contact the authors of this review.

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
