# Peer review of "Opinions of Older Individuals on Advance Care Planning and Factors Affecting Their Views: A Systematic Review"

_ijerph, 2023, doi:10.3390/ijerph20105780_

Round 1

Reviewer 1 Report

I thank the authors for their paper and their research!

The presented topic is very delicate and this review could be useful in order to better evaluate the problem and, in my opinion, to deeply reason about its complexity. 

I consider the work well done and very well organized in order to put in evidence the different details of the analysis. Nevertheless, I think, as the same authors say, that  referring to just 17 papers is too limiting for the value of the paper...but it is a first important step in any case.

If I've not misunderstood, in lines 295-299, I would suggest to better declare which are the 7+6+9 papers quoted...and moreover to spend some more word about the reference to the percentage of male and female, if you have from the original papers, because it is just a sentence in the whole paper.

Here some more remarks essentially related to the conclusions and the possible further researches:

1) It is not mentioned any reference to the importance of the preparation to death along the course of the life, according to a possibile concrete and reasonable religious faith able to give meaning to life itself;

2) there is a marginal reference to the complexity and the importance of the palliative care, as a cross-disciplinary integral approach to the person in suffering. The time of palliative care could be a very precious time both for families and for the sicks; probably also from this point of view could be collected further elements;

3) it is never mentioned the spiritual and religious assistance as a decisive horizon for dealing with the meaning of life, of death and of care; the religious references seems to be marginal, in a topic (ACP) which cannot be considered just in an horizontal persepctive, because touches the mistery of life;

4) there is no distinction between spirituality - religiosity and faith which are three general pillars in their differences and in their contents, also important in relation to the approach to care planning;

As you can see, these remarks are not intended to revise the document, but would only like to offer some more perspectives for research.

Thank you.

Author Response

The presented topic is very delicate and this review could be useful in order to better evaluate the problem and, in my opinion, to deeply reason about its complexity. 

I consider the work well done and very well organized in order to put in evidence the different details of the analysis. Nevertheless, I think, as the same authors say, that  referring to just 17 papers is too limiting for the value of the paper...but it is a first important step in any case.

If I've not misunderstood, in lines 295-299, I would suggest to better declare which are the 7+6+9 papers quoted...and moreover to spend some more word about the reference to the percentage of male and female, if you have from the original papers, because it is just a sentence in the whole paper.

Authors: The changes have been added to this part.

Here some more remarks essentially related to the conclusions and the possible further researches:

1) It is not mentioned any reference to the importance of the preparation to death along the course of the life, according to a possibile concrete and reasonable religious faith able to give meaning to life itself;

2) there is a marginal reference to the complexity and the importance of the palliative care, as a cross-disciplinary integral approach to the person in suffering. The time of palliative care could be a very precious time both for families and for the sicks; probably also from this point of view could be collected further elements;

3) it is never mentioned the spiritual and religious assistance as a decisive horizon for dealing with the meaning of life, of death and of care; the religious references seems to be marginal, in a topic (ACP) which cannot be considered just in an horizontal persepctive, because touches the mistery of life;

Authors: the religious characteristics were evaluated in some researches of the data and we emphasised this.

4) there is no distinction between spirituality - religiosity and faith which are three general pillars in their differences and in their contents, also important in relation to the approach to care planning;

Authors: Religion and religious believes were indicated in the some articles of our data. But they did not focus on distinction between spirituality, faith and religiosity. We agree that should be considered for further studies.

As you can see, these remarks are not intended to revise the document, but would only like to offer some more perspectives for research.

Thank you.

Authors: We appreciated these all suggestions that are so valuable. However we did not focus on specifically on religious behavior or spirituality on ACP. We will consider these suggestions for further research as the reviewer has already indicated that.

Reviewer 2 Report

The authors submitted a manuscript on advance care planning and older individuals’ relevant views.

It is not clear how the methodological approach has been designed and how it fits a scope.

The major point is the lack of consensus and understanding of palliative care and old age.

Both terms are not clear or set according to universal standards, and the picture is getting more complicated because of the disparate national policies and the vague application of criteria.

Defining old age has received severe criticism, but in no case refers to people in the 50-65 years old range. People >50 could be considered older adults. I have no objection against the criterion choice, but revisions have to be applied all over. Do specify the sample demographics, whether there have been diagnosed with a disease, and whether ACP was mandatory.

Accordingly, this methodological restriction is interrelated to palliative care, as advanced care planning requires a person to make an informed decision. It is possible for someone to make a decision prior to any diagnosis, but it is probable to proceed to ACP or revise ACP when a condition arises, and most important when experiencing symptoms and care. Life and life factors are dynamic, and this is supported by people responses.

Therefore, it is crucial to understand that specific groups may be valuable in understanding the view shift upon diagnosis and understand the barriers and facilitators. This understanding has been reached. Please advise the ACP conversations section and the highlighted recommendations from Advance care planning in dementia: recommendations for healthcare professionals, 10.1186/s12904-018-0332-2 .

Overall, there is a feeling that the main effort has been put in shorting the data. There are descriptive paragraphs. However, the initial and final parts are fragmented and do not address a contemporary synthesis and analysis of the findings in a written-narrative format.

Regarding the “quantitative” studies it is not clear how they could improve our knowledge.

I would advise the authors to read

https://bmcmedresmethodol.biomedcentral.com/articles/10.1186/s12874-020-0898-2

https://pubmed.ncbi.nlm.nih.gov/30089228/

to explore their options and handle the narrative process, and/or apply another tool.

L22

Correct to MasterFILE

Is it TR index or TR Dizin?

L35-L41

Modern medicine is acknowledging palliative care and such approaches are occasionally (?) implemented.

Please advise WHO resources regarding the scope of palliative care and the approaches. Then introduce and fully describe the value of ACP.

A reference has already been included #7 [Definition and recommendations for advance care planning: an international consensus supported by the european association for palliative care]. Please advise carefully.

L39

“However, they might fail to respond to orders’ emotional needs”

The meaning is unclear.

In addition, the authors mention searching for literature in Greek and Turkish.

The book is in German.

L126-127

The MMAT uses a series of questions in which four possible answers specific to the study design are converted into binary scores (from the lowest to the highest: 25 points/ 50 points/ 75 points/ 100 points).

It is not clear how the four scales are transformed into binary, 0 vs 1, meaning 0<50 and 1>50 or what?

L496

In according?

On line and off line literature format needs to be revised, e.g. capitals when necessary for #7, reference to page (L61-62) etc.

Author Response

It is not clear how the methodological approach has been designed and how it fits a scope.

The major point is the lack of consensus and understanding of palliative care and old age.

Both terms are not clear or set according to universal standards, and the picture is getting more complicated because of the disparate national policies and the vague application of criteria.

Authors: The term ACP is the main concept of the study. We do not focus on palliative care concept.

Defining old age has received severe criticism, but in no case refers to people in the 50-65 years old range. People >50 could be considered older adults. I have no objection against the criterion choice, but revisions have to be applied all over. Do specify the sample demographics, whether there have been diagnosed with a disease, and whether ACP was mandatory.

Authors: We indicated the age over 50 in the inclusion criteria and we did not add the diagnosed disease and ACP conditions to inclusion criteria. We gave demographics features of the samples in the Table1.

Accordingly, this methodological restriction is interrelated to palliative care, as advanced care planning requires a person to make an informed decision. It is possible for someone to make a decision prior to any diagnosis, but it is probable to proceed to ACP or revise ACP when a condition arises, and most important when experiencing symptoms and care. Life and life factors are dynamic, and this is supported by people responses.

Therefore, it is crucial to understand that specific groups may be valuable in understanding the view shift upon diagnosis and understand the barriers and facilitators. This understanding has been reached. Please advise the ACP conversations section and the highlighted recommendations from Advance care planning in dementia: recommendations for healthcare professionals, 10.1186/s12904-018-0332-2 .

Authors: We believe that this suggestion will give a valued contribution to our paper. We added this idea to the conclusion part.

Overall, there is a feeling that the main effort has been put in shorting the data. There are descriptive paragraphs. However, the initial and final parts are fragmented and do not address a contemporary synthesis and analysis of the findings in a written-narrative format.

Regarding the “quantitative” studies it is not clear how they could improve our knowledge.

I would advise the authors to read

https://bmcmedresmethodol.biomedcentral.com/articles/10.1186/s12874-020-0898-2

https://pubmed.ncbi.nlm.nih.gov/30089228/

to explore their options and handle the narrative process, and/or apply another tool.

Authors: In the research, we seek to answer the research questions “How do older individuals approach the ACP process?” and “What are the factors that affect individuals' views of ACP?” We chose narrative analysis because of the diverse methods of the researches and our research questions according to the aim of the study. In the first part of the findings, we analyzed selected qualitative and mixed researches according to the research question and got theoretical themes and according to these themes, we gave some quotes from the articles to explain these themes. In the second part of the findings we tried to answer the second research question “What are the factors that affect individuals' views of ACP?” and the answers were discriptive because of the aim of the study and research question.

We appreciated the suggestion of the reviewer, however, it is not possible to change the initial and final parts. We consider these suggestions for further studies.

L22

Correct to MasterFILE

Is it TR index or TR Dizin?

Authors: It has been corrected.

L35-L41

Modern medicine is acknowledging palliative care and such approaches are occasionally (?) implemented.

Authors: It has beed changed.

Please advise WHO resources regarding the scope of palliative care and the approaches. Then introduce and fully describe the value of ACP.

Authors: It has beed changed and described the value of ACP.

A reference has already been included #7 [Definition and recommendations for advance care planning: an international consensus supported by the european association for palliative care]. Please advise carefully.

L39

“However, they might fail to respond to orders’ emotional needs”

The meaning is unclear.

In addition, the authors mention searching for literature in Greek and Turkish.

The book is in German.

Authors: It has been corrected. The reference is in German, we searched the studies in English and Turkish.

L126-127

The MMAT uses a series of questions in which four possible answers specific to the study design are converted into binary scores (from the lowest to the highest: 25 points/ 50 points/ 75 points/ 100 points).

It is not clear how the four scales are transformed into binary, 0 vs 1, meaning 0<50 and 1>50 or what?

 Authors: It has been corrected.

L496

In according?

Authors: It has been added

On line and off line literature format needs to be revised, e.g. capitals when necessary for #7, reference to page (L61-62) etc.

Authors: It has been corrected.

Reviewer 3 Report

This systematic review focuses on older individuals' views on advance care planning (ACP) and the factors that influence those opinions. The review analyzed 20 search terms in databases over the last 10 years and included studies with individuals aged 50 and over but excluded articles with a specific disease sample. The results showed that older adults' lack of knowledge and awareness played a critical role in their participation in ACP-related processes. Variables affecting their views included advanced age, marital status, socioeconomic status, perception of remaining life expectancy, self-perceived health, number and stage of chronic diseases, religion, and cultural characteristics. The study suggests that disseminating knowledge and awareness of ACP is necessary and that individual characteristics and experiences should be considered in training and awareness studies. The review's strength lies in the fact that both qualitative and quantitative data on ACP were examined, and the views of individuals and the factors affecting those views are presented. The limitation of the study is that it only searched for studies published in English and Turkish in the last 10 years, potentially missing relevant studies. The review also found that individuals with higher socio-economic status and education levels were more knowledgeable about ACP and more likely to participate in ACP. Moreover, those with previous ACP experience were found to have a more positive view of ACP participation and discussion. Other factors influencing older adults' attitudes towards ACP included race, family relationships, confidence in the future, and religious beliefs. The review concludes that efforts to increase public knowledge and awareness of ACP should consider individual socio-economic status, educational status, chronic diseases, and religious beliefs. One limitation of the review is the limited scope of the search, which was only limited to two languages and a 10-year period. This could have potentially missed relevant studies on the topic. However, the review's strength lies in its analysis of both qualitative and quantitative data and its inclusion of individual views and factors affecting those views.

This review provides valuable insights into the factors that influence older adults' attitudes towards ACP. This knowledge can help healthcare professionals and policymakers develop effective strategies for promoting ACP and improving end-of-life care for older adults.

The article is well written and is suitable for the publication in this journal.

Author Response

Thank you for your encouraging and supportive review.  

Reviewer 4 Report

This is an interesting and thought-provoking study, and I have only a few minor comments. The study divides subjects into religious and non-religious; is it possible to define these groups more explicitly.  Not all religions are the same, and I would have thought there is some variation in the interpretation of how to balance between relief of suffering and the prolongation of life. Although, the need for ACP would not be precluded wherever the patient feels the balance should lie, I think this issue should be addressed.

In my experience a major impediment to ACP is when a patient or their family is from a different cultural and/or religious background, when there is often misinterpretation of what is being suggested and its underlying motivation. It does not seem that this aspect has been addressed. However, I note that some studies specifically looked at immigrant populations, such as Chinese patients in the USA, which suggests that this issue might have been why the study was performed.

Future time perspective is an important concept that not all readers may be familiar with. I suggest that is more fully explained and defined.

Minor issues:

1. The abbreviation ACP appears in the text before it is defined.

2. The following sentences in the Introduction are not clear:

“These can prolong life until death while also posing a significant challenge for older individuals and their relatives. However, they might fail to respond to orders’ emotional needs [1].”

3. The main findings of Amjad et al (Table) are not clear:

“Of 304 participants, 84% had one or more personal experiences or experience with others. Personal experiences were not associated with greater readiness for most ACP behaviours.”

Author Response

This is an interesting and thought-provoking study, and I have only a few minor comments. The study divides subjects into religious and non-religious; is it possible to define these groups more explicitly.  Not all religions are the same, and I would have thought there is some variation in the interpretation of how to balance between relief of suffering and the prolongation of life. Although, the need for ACP would not be precluded wherever the patient feels the balance should lie, I think this issue should be addressed.

In my experience a major impediment to ACP is when a patient or their family is from a different cultural and/or religious background, when there is often misinterpretation of what is being suggested and its underlying motivation. It does not seem that this aspect has been addressed. However, I note that some studies specifically looked at immigrant populations, such as Chinese patients in the USA, which suggests that this issue might have been why the study was performed.

Authors: We appreciated  for all suggestions of the reviewer It has been added to the discussion part, shortly.

Future time perspective is an important concept that not all readers may be familiar with. I suggest that is more fully explained and defined.

Authors: It has been added.

 Minor issues:

  1. The abbreviation ACP appears in the text before it is defined.

Authors: The definition was added before ACP abbreviation.

  1. The following sentences in the Introduction are not clear:

“These can prolong life until death while also posing a significant challenge for older individuals and their relatives. However, they might fail to respond to orders’ emotional needs [1].”

Authors: It has been changed.

  1. The main findings of Amjad et al (Table) are not clear:

“Of 304 participants, 84% had one or more personal experiences or experience with others. Personal experiences were not associated with greater readiness for most ACP behaviours.”

Authors: It has been changed.